# CHAIN-OF-THOUGHT REASONING IS A POLICY IMPROVEMENT OPERATOR

> "Self-education is, I firmly believe, the only kind of education there is."
>
> *Isaac Asimov*

## ABSTRACT

Large language models have astounded the world with fascinating new capabilities. However, they currently lack the ability to teach themselves new skills, relying instead on large amounts of human-generated training data. We introduce *SECToR* (**S**elf-**E**ducation via **C**hain-of-**Th**ought **R**easoning), a proof-of-concept demonstration that language models can teach themselves new skills using chain-of-thought reasoning. During the self-learning loop, SECToR asks models to solve addition problems using chain-of-thought reasoning before training the next version of the model to solve those same problems directly *without using such reasoning*. This process often results in an improved model which can, when again augmented with chain-of-thought reasoning, solve even harder problems than the original model, allowing the self-learning loop to continue. Language models trained via SECToR autonomously learn to add up to 29-digit numbers without access to any ground truth examples beyond an initial supervised fine-tuning phase consisting only of numbers with 6 or fewer digits. Our central hypothesis is that chain-of-thought reasoning can act as a policy improvement operator, similarly to how Monte-Carlo Tree Search is used in AlphaZero (Silver et al., 2017). We hope that this research can lead to new directions in which language models can learn to teach themselves without the need for human demonstrations.

## 1 INTRODUCTION

Large language models are currently trained on vast corpora of human-generated data (Vaswani et al., 2023; Devlin et al., 2019; Radford et al., 2019; Brown et al., 2020; Chowdhery et al., 2022; Touvron et al., 2023). While large language models have demonstrated many surprising capabilities, the possibility of reaching superhuman performance is a challenging proposition when training solely on existing data. In this paper, we ask whether large language models can autonomously teach themselves new skills rather than solely depending on the availability of suitable data. A positive answer to this question would open the door to a tantalizing possibility. Although the discovery of scaling laws (Kaplan et al., 2020; Hoffmann et al., 2022) for language models has created much excitement for training increasingly larger language models, these models have already consumed a significant fraction of the high-quality (textual) data on the internet. The issue of data exhaustion is an active area of research (Villalobos et al., 2022), especially in light of results showing that repeated training on the same data quickly leads to degeneration (Shumailov et al., 2023). If large language models can effectively learn from data they themselves generate, this could usher in a new era of scaling laws that are solely compute-driven, independent of how much data is available.

Self-training is not a novel concept in AI. For instance, AlphaZero achieved superhuman capabilities in Go, Chess, and Shogi via self-play (Silver et al., 2017). Nevertheless, success with such a process has not yet been demonstrated with language models. Recently, Wei et al. (2022) highlighted the intriguing observation that language models often produce better results when prompted to use *chain-of-thought reasoning* to solve the problem. This approach contrasts with the conven-

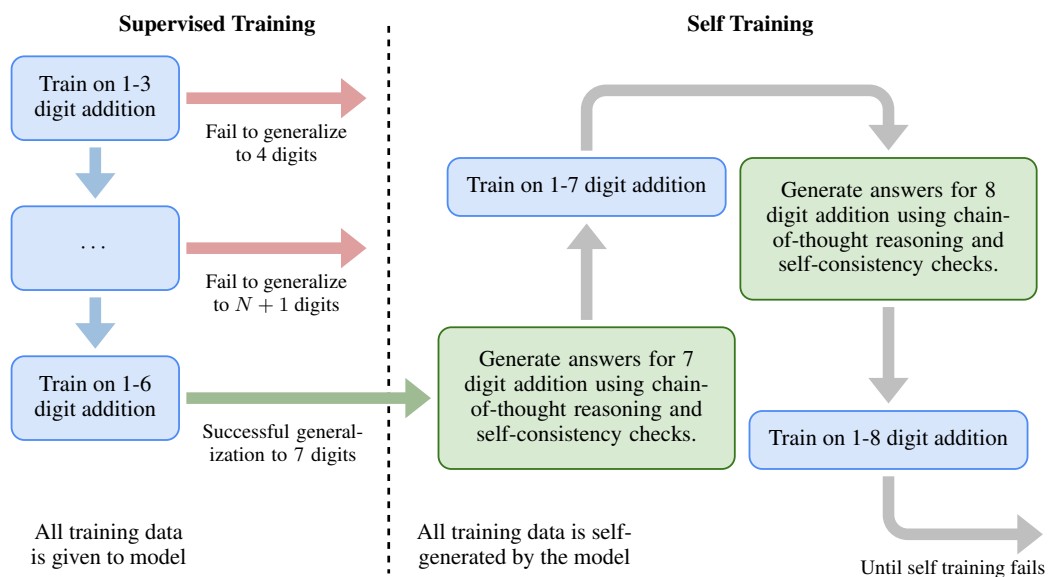

Figure 1: SECToR begins with a supervised fine-tuning phase in which the model is trained with curriculum learning in which harder addition problems after the model has mastered solving easier ones. SECToR begins self-training when the models generalize perfectly to $N + 1$ digit addition despite only have been trained on 1 through $N$ digit addition thus far when equipped with chain-of-thought reasoning. Empirically, this occurred at 7 digits for the 582M parameter model. During self-training, training is largely identical to the supervised training phase, except that all training data is generated by the model itself using chain-of-thought reasoning and self-consistency checks without any external verification of correctness (see Figure 12).

tional method where models are prompted to instantly generate an answer, without producing any intermediate steps.

In this paper, we introduce *SECToR* (**S**elf-**E**ducation via **C**hain-of-**Th**ought **R**easoning), which gives a proof-of-concept that large language models can successfully teach themselves new abilities via chain-of-thought reasoning, using addition as a benchmark task. Despite well-known difficulties for language models to perform addition (Liu & Low, 2023; Nye et al., 2021; Lee et al., 2023), language models trained via SECToR teach themselves to add numbers with up to 29-digits *without access to any ground-truth examples* for numbers of length longer than 6 digits.

The observation that lies at the heart of SECToR is that chain-of-thought reasoning can be considered a policy improvement operator: *regardless of the quality* of the underlying model (assuming it satisfies some minimum size and training requirements), models that are prompted to use chain-of-thought reasoning outperform directly sampling an answer from the model. SECToR uses this observation to perform self-learning. During self-learning, SECToR prompts the model to use chain-of-thought reasoning to generate solutions to problems that it could not otherwise solve without such reasoning. Then, this same model is fine-tuned to generate these exact solutions, this time *without using chain-of-thought reasoning*. This leads to an improved version of the model which can now directly solve problems that the previous version of the model required using chain-of-thought reasoning to solve. If this improved model, when again equipped with chain-of-thought reasoning, can solve a larger set of problems than the original model could (even when equipped with chain-of-thought reasoning), the self-learning process can continue. This procedure is highly similar to the procedure that AlphaZero (Silver et al., 2017) used to reach superhuman performance in Chess, Go, and Shogi, except using chain-of-thought reasoning in place of Monte-Carlo Tree Search as the policy improvement operator.

In the specific task of addition, SECToR begins with a pre-trained language model and then performs an initial supervised fine-tuning period in which it learns to perform addition both with and without using chain of thought reasoning but only for addition problems with a small number of digits. After

the initial supervised training phase, it then undergoes a self-training phase in which all training data is model-generated, *with zero access to ground-truth data*. During each learning period, we train the language model digit-by-digit using curriculum learning by requiring the model to perform satisfactorily at 1 to $N$ digit addition before introducing $N + 1$ digit problems to the dataset. Examples of each of these kinds of tasks are shown in Figure 2. This setup is analogous to prior work such as AlphaGo (Silver et al., 2016), where the model was initially trained on human-generated data before undergoing self-training.

| Task Type | Example |
|-----------|---------|
| Addition w/o CoT (fast) | Q: 141 + 123 = ? A: 264. |
| Addition w/ CoT (slow) | Q: 141 + 123 = ? A: The first number's last digit is 1. The second number's last digit is 3. 1 + 3 = 4. The last digit of this sum is 4 so our initial partial answer is 4. We can now recursively solve a smaller problem. Removing the last digit from each number we have 14 and 12. The next subproblem is 14 + 12. |

Figure 2: Examples of the two types of addition problems. One can view these two types as the analogues of Type 1 (fast) and Type 2 (slow) reasoning (Kahneman, 2011).

A major challenge that has prevented past efforts of self-learning in language models from succeeding, especially in arithmetic, is a phenomenon that we call *error avalanching*. During self-training, when all training data is generated by the model itself, there is no guarantee that the data is correct. *Error avalanching* occurs when small errors or inaccuracies in a model's output compounds rapidly over time, because each iteration of the model learns from the outputs of the previous model amplifies the existing mistakes. If left unchecked, error avalanching leads to severe degradation of performance within only a few iterations of self-training (see Figure 4). This is consistent with past attempts to get language models to self-learn (for addition as well as other tasks) in which improvement stagnates in only a few steps at most (Zelikman et al., 2022; Lewkowycz et al., 2022; Bai et al., 2022; Huang et al., 2022; Jung et al., 2023). While error avalanching is a fundamental issue in any bootstrapped process, SECToR manages to largely mitigate error avalanching via several forms of self-consistency checks (Figures 4 and 5) that minimize the number of mistakes introduced to the dataset. Nevertheless, SECToR does not continue *ad infinitum*, and training eventually terminates due to accumulated errors. We return to this issue in the discussion.

**Results.** Our results indicate that the pre-trained 582M parameter ByT5 model (Xue et al., 2022), after a supervised fine-tuning period that only provides examples for 1 to 6 digits, can teach itself to perform up to 29-digit addition with 98+% accuracy – successfully undergoing 22 steps of self-improvement in the process (Table 1). Additionally, a smaller training run with the 300M parameter version of ByT5 showed similar positive results, successfully teaching itself to perform up to 24-digit addition after a supervised learning phase that included examples of up to 8 digits (Appendix D).

## 2 APPROACH

### 2.1 REASONING AS A POLICY IMPROVEMENT OPERATOR

The concept of a policy improvement operator plays a central role in reinforcement learning. In reinforcement learning, a policy is the strategy by which an agent chooses its actions, given the current state of the world. A policy improvement operator is a function that takes in an arbitrary policy and returns an improved policy, which is closer to the optimal policy, with respect to a given reward function. Many reinforcement learning algorithms, such as Q-learning or policy gradients, are built around a policy improvement operator. A crucial property of a policy improvement operator is that it can be used repeatedly —— if one continually applies a policy improvement operator to a policy, it will eventually converge to the environment's optimal policy.

In AlphaZero, Monte-Carlo Tree Search (MCTS) served as the policy improvement operator. Nevertheless, two-player zero-sum games are relatively simple, well-defined environments. In this paper, we explore the hypothesis that *reasoning can serve as a policy improvement operator*, analogous

to how MCTS functions in AlphaZero, but for a broader range of environments. To complete the analogy, we consider the language model's conditional distribution over next token as the "policy," the previous context as the "environment." The goal is for each version of the model to generate the data upon which the next version of the model is trained using the policy improvement operator (reasoning).

While large language models have not yet been shown to possess a complete control of general reasoning skills, Wei et al. (2022) showed that a method called "chain-of-thought reasoning" could generate substantial performance improvements merely by asking models to think through a problem step-by-step. In particular, Wei et al. (2022) noticed that often a model *unable to solve a problem* without using reasoning was able to come up with the correct answer if allowed to reason through the problem step-by step. One perspective on chain-of-thought reasoning is that the model is spending computational resources at inference time to augment its own abilities. This is similar to how AlphaZero spends compute to perform MCTS and augment its own game playing abilities. Given that chain-of-thought reasoning can allow models to solve problems that they otherwise could not, it is natural to ask whether repeated application of this method might allow models to self-improve far beyond their original capabilities.

At a high level, SECToR uses chain-of-thought reasoning as a policy improvement operator to assist the model in solving problems that it could not do without using the additional computation. It then constructs a dataset of these new problem, solution pairs, which is then use to train the model to solve these same problems *without using chain-of-thought reasoning.* The hope is that, having learned to directly solve problems that the previous version of the model could only tackle with the additional computational power of chain-of-thought reasoning, this improved model will be able to solve an even larger set of problems when again augmented with chain-of-thought reasoning.

## 2.2 Error Avalanching

The approach of repeatedly training the model on its own utterances usually quickly results in a phenomenon we call error avalanching. Consider a scenario where the model makes a small error in one iteration. When this output is fed back into the model for the next round of training, the error becomes part of the training data. As the model learns from this flawed data, the error may be reinforced rather than corrected. In subsequent iterations, this error may compound leading to increasingly inaccurate outputs. This is akin to an avalanche, where a small disturbance can trigger a massive slide.

Because we ask the model to learn without giving it any access to the ground truth, it is important to prevent this snowballing of errors, eventually derailing the training process. In past attempts to use similar self-learning approaches, this phenomenon caused the training process to go awry within just a few steps (Zelikman et al., 2022; Jung et al., 2023; Bai et al., 2022; Huang et al., 2022). In order to mitigate the effects of error avalanching, SECToR utilizes several consistency checks largely based on the following idea: we ask the model to generate a large number of "equivalent" variations of any given question and then ask the model to (independently) solve each variation. If the answers are largely consistent, we accept these answers into the training data. Otherwise, SECToR discards the answers. These consistency checks are essential for SECToR to minimize the introduction of incorrect data into the model's self-training loop.

## 3 Experiments

### 3.1 Setup

Addition is a fundamental task in mathematics, but one on which language models have historically struggled to perform. As such, we use it as a toy dataset for evaluating self-learning. As addition can be considered a simple form of problem-solving, a language model being able to teach itself addition may indicate potential for applying general logic and reasoning to solve more complex problems.

We ask whether models can teach themselves how to add very large numbers after having been shown only addition examples with substantially smaller numbers. In our setup, we consider only problems of the form $a + b$ or $a + b + 1$, where $a$ and $b$ are guaranteed to have the same number of digits. $a$ and $b$ are uniformly sampled among all integers with the proper number of digits. We train

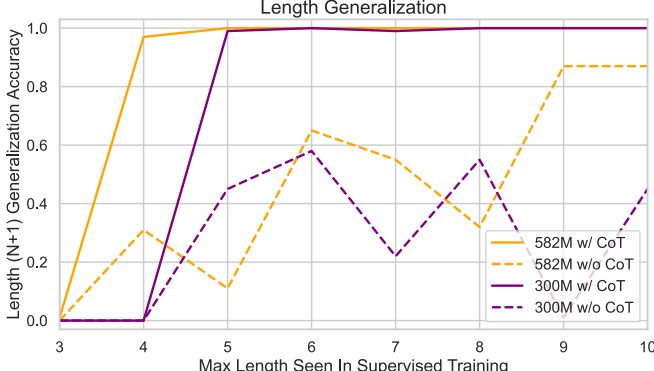

Figure 3: This figure describes the generalization accuracy of models to $N + 1$ digit addition when they have only been trained on 1 through $N$ digit addition. Consistent with past work, we find that models typically show poor length generalization when asked to perform standard (fast) addition. However, if we allow the models to use chain-of-thought reasoning, we find that the generalization accuracy reaches almost perfect accuracy after training on only a few digits.

all models on a cluster of 8 V100 GPUs with 32 GB of memory each. Prior work has indicated the importance of proper tokenization for performing arithmetic (Liu & Low, 2023; Zhou et al., 2022). In order to avoid any such issues, we use the ByT5 family, which operates on the raw character/byte value. A detailed description of the hyperparameters used in training can be found in Appendix B.

### 3.2 SUPERVISED TRAINING

Without fine tuning, we find that the ByT5 models are quite poor at addition (Appendix C), which is consistent with prior work on benchmarking addition. SECToR thus begins by performing an initial supervised fine-tuning phase consisting of addition problems with only a small number of digits before beginning the self-training loop. This process is described in Figure 1.

During this supervised training period, models are trained to perform addition both with and without chain-of-thought reasoning. In contrast to the self-learning phase of training, all training examples are generated programmatically by an external script during supervised learning. "Fast" addition consists of asking models to immediately output the solution to the addition problem without using any intermediate tokens to reason through the answer. In contrast, "slow" addition consists of asking models to do a single simplification step of turning an $N$ digit problem into an $N - 1$ digit problem and a partial solution, similar to the method of teaching addition to schoolchildren. In this setup, performing "slow" addition refers do performing a single step of simpliciation, not fully solving the problem. Figure 2 depicts examples of both "fast" and "slow" addition. Training examples for each type are prefixed with a special token depicting their type, so the model can differentiate between the two. Learning both how to add numbers directly as well as via chain-of-thought addition can be viewed as a similar idea to that of *process supervision* (Lightman et al., 2023), which recently achieved state-of-the-art performance on the MATH dataset (Hendrycks et al., 2021).

The supervised training phase follows a curriculum learning schedule where a model must first achieve sufficient accuracy on 1 through $N$ digit addition before $N + 1$ digit examples are added to the dataset, starting with $N = 3$. Accuracy is measured by computing an exact token match between the gold reference text and the model output while sampling at temperature 0. Any answer that is not in the correct format is automatically marked as incorrect. To prevent catastrophic forgetting (McCloskey & Cohen, 1989), we mix examples from 1 through $N$ digit addition while training the model on $N + 1$ digit addition. Details of the precise composition of training examples are provided in Section B. In Appendix J, we run an ablation to measure the effects of the curriculum as compared to simply doing a single supervised tuning phase where we learn 1 to $N$ digit addition jointly.

The supervised phase of training concludes when models exhibit length generalization to $N + 1$ digit addition problems when performing "slow", chain-of-thought augmented addition. In Appendix I, we describe experiments regarding emergent properties of language models, suggesting that larger models need a shorter supervised training period before exhibiting such generalization. This leads to a hypothesis that a sufficiently large pre-trained language model might be able to forgo the supervised training period entirely and begin self-training immediately, perhaps with only a few examples of in-context demonstrations.

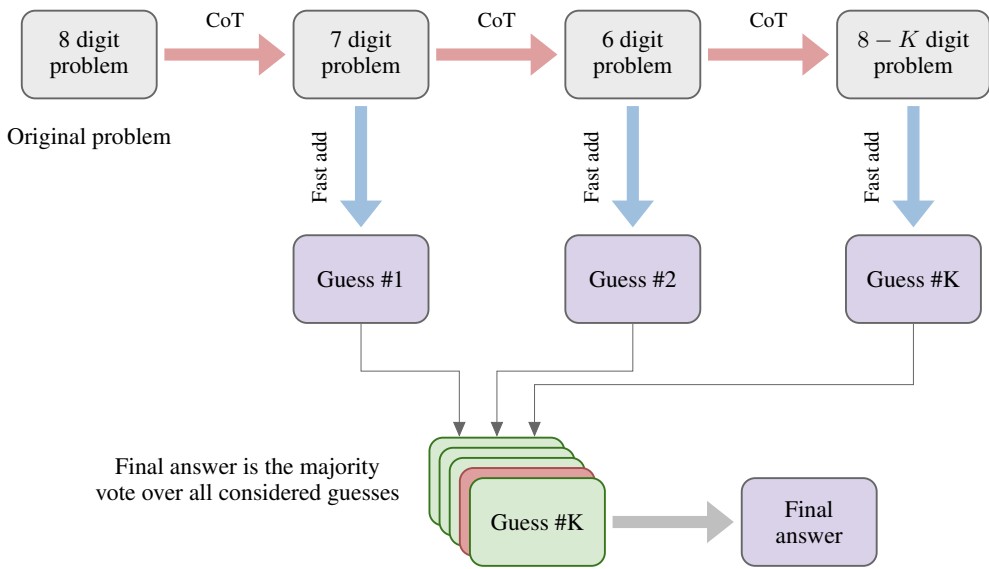

Figure 4: Simplify-then-guess asks the model to simplify the problem $K$ times. After each simplification, the model directly guesses the final solution without any further reasoning steps. These guesses are then aggregated via majority vote to produce the final answer. Simplify-then-guess allows the model to balance the generalization ability of slow, chain-of-thought reasoning, with a consistency check of having $K$ separate guesses of the answer.

## 3.3 SELF-TRAINING

The second period of learning consists of self-training. The distinguishing feature of self-learning is that *all new training data in this phase is generated by the model itself without any external verification of correctness.* Aside from this important difference, self-training largely follows a similar setup to that of the supervised learning period. Here, the goal is to ask whether the model can continue to teach itself skills even in the absence of human demonstrations, as is often the case in reinforcement learning setups. However, unlike standard reinforcement learning, SECToR does not give the model the privilege of querying the environment: models must learn entirely based on their own conceptions of the environment without any grounding in the real world.

### 3.3.1 GENERATING NEW ADDITION EXAMPLES

The self-training phase requires the model being able to generate both "fast" and "slow" styles of addition for numbers larger than it has seen in training. This is possible largely due to the observation that the reasoning capabilities of models generalize exceptionally well to longer addition lengths beyond what was seen during training. Figure 3 shows that the 582M parameter model begins to generalize to novel digit lengths after training only on 1 through 4 digit addition and reaches almost perfect generalization accuracy to $N + 1$ digit addition after training on 1 through 6 digit addition. Similar results can be obtained for the 300M parameter model, albeit with a longer supervised training period (see Appendix I for a longer discussion on emergence). This generalization capabilities form the heart of how language models are able to perform self-learning with SECToR.

As described in Section 3.2, the curriculum learning setup requires that models successfully learn both fast and slow styles of 1 through $N$ digit addition before engaging in learning $N + 1$ digit addition. Additionally, the precondition for ending the supervised phase of learning and beginning self-training is that models achieve near-perfect generalization accuracy on slow, chain-of-thought augmented $N + 1$ digit addition. This precondition immediately suggests a method of generating training examples for "slow" $N + 1$ digit addition: simply directly sampling from the model using greedy decoding. Generating "fast" style training examples is a more complex process. Figure 3 shows that longer after models generalize using chain-of-thought, generalization without such reasoning remains poor. To generate solutions to $N + 1$ digit addition problems, SECToR uses a

novel decoding method called *simplify-then-guess* which utilizes a model's abilities to perform both fast and slow addition for 1 through $N$ digit addition (Figure 4). *Simplify-then-guess* is inspired by the approaches of least-to-most prompting (Zhou et al., 2023) and self-consistency (Wang et al., 2023b). In least-to-most prompting, models are prompted to decompose problems into simpler sub-problems before solving each one in sequence. Simplify-then-guess follows a similar process, but adds in a built in self-consistency check inspired by Wang et al. (2023b). After each simplification of the problem, simplify-then-guess asks the model to directly guess the final solution without using any further reasoning steps. The final answer is a majority vote over all intermediate guesses. For example, if a model is tasked with solving an 8-digit addition problem, it will first simplify the problem into a 7 digit addition problem before taking its first guess of a solution. It then repeats this process with the 7-digit problem it just generated, and so on. This process is described in Figure 4.

The primary advantage of simplify-then-guess over least-to-most prompting, is that with least-to-most prompting, a single error at any point in the process corrupts the entire solution. In contrast, simplify-then-guess has a built-in error check in that, because the accuracy of each "guess" is unaffected by any reasoning errors that occur after the guess is made. This error reduction is critical for mitigating error avalanching. In SECToR, simplify-then-guess generates $K$ separate guesses for an addition problem by applying between 1 and $K$ simplification steps before fast adding the remaining addition problem. It then takes a majority vote between the generated guesses to construct a final guess for the answer. In this paper, we use $K = 5$, which was found to be a good balance between computational speed and accuracy.

### 3.4 COMMUTATIVITY CHECKS

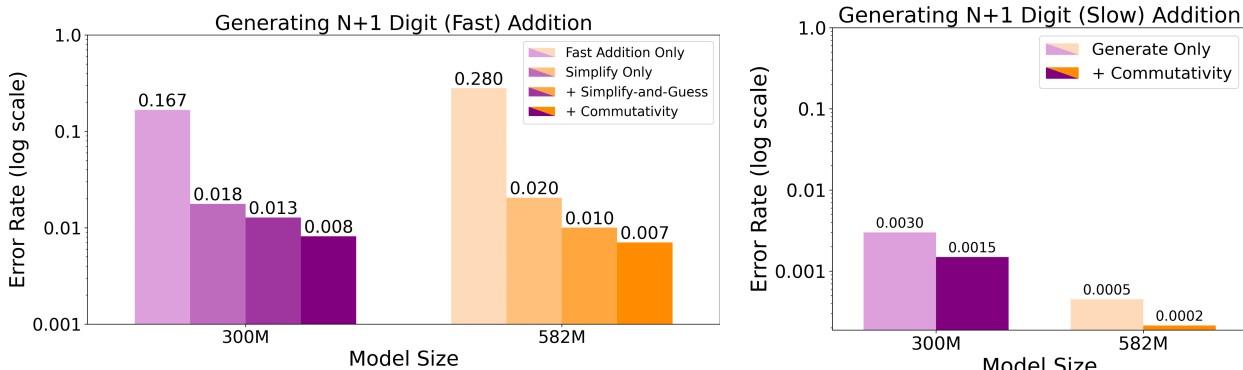

Figure 5: Generating training data for $N + 1$ digit "fast" addition via directly sampling from the model (via Fast Addition) leads to extremely high error rates. Generating training data purely using chain-of-thought reasoning to simplify a problem (Simplify Only) fares better, but not as well as simplify-then-guess, which utilizes a model's ability to perform both fast and slow types of addition. This error is then further reduced by a self-consistency check based on commutativity. Error rate when generating new "slow" addition training examples is substantially lower, both before and after using the commutativity self-consistency check due to the observation that models exhibit strong length generalization when equipped with chain-of-thought reasoning.

Nevertheless, simplify-then-guess alone is not enough to mitigate error avalanching. We perform an additional self-consistency check based on commutativity. Specifically, we ask the model to solve independently (via simplify-then-guess) both a problem $a + b$ and its twin $b + a$. If the generated solutions are not equal, we discard these problems from the dataset. For fast-type addition problems, this check requires that a problem and its twin have answers that are identical, as measured by an exact string match. For slow-type addition problems, this commutativity check is slightly more difficult due to the observation that the chain-of-thought reasoning utterance for $A + B$ is not the same as the answer for $B + A$. Instead, SECToR checks that the answers generated by simplifying for one step, followed by immediately fast adding the subproblem emit identical answers for both a problem and its commutative twin.

### 3.5 RESULTS

We report the results for a single training run for the 582M parameter model. A similar training run with the 300M parameter model in described in Appendix D. Additional replication experiments are included in Appendix E. The 582M model began self training after 6 digits (Figure 10). Self learning terminated after successfully learning up to 28 digit addition problems, having failed to successfully learn continue the training loop with 29 digit addition. Figure 6 describes the generalization accuracy of the 582M parameter model over the course of training. Table 1 reports the addition accuracies achieved by the final version of the model. We conduct a more careful analysis errors of the model in Appendix H.

| Length | 1-29 | 30 | 31 | 32 | 33 | 34 | 35+ |
|---|---|---|---|---|---|---|---|
| **Accuracy (%)** | 98-100 | 88 | 79 | 52 | 26 | 2 | 0 |

Table 1: Accuracy (out of 100 examples) of the final checkpoint of the 582M model after training. For example, this table shows that the post-training 582M model can add 30 digit numbers with 88% accuracy without using any chain-of-thought reasoning.

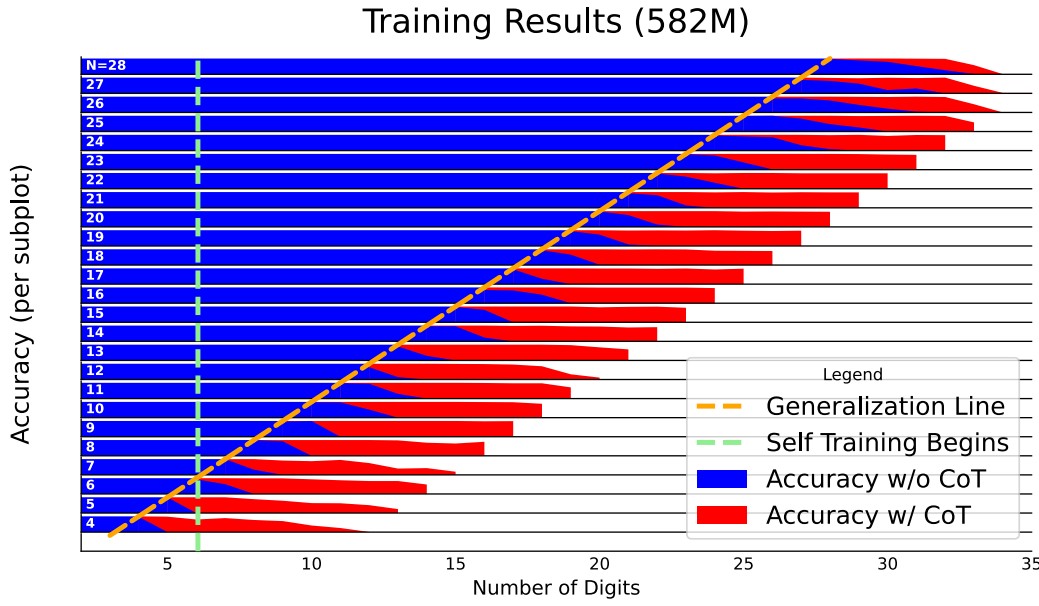

Figure 6: This figure describes the training run of the 582M parameter model. Each row label $N$ denotes a model which has completed training on 1 through $N$ digit addition. The shaded regions depict a model's generalization accuracy beyond $N$-digits, measured up to a maximum of $N + 8$ digits. Blue represents accuracy without using chain-of-thought reasoning, while the red allows it. Model accuracy both with and without chain-of-thought continues to grow after self-training begins at 7 digits, despite no additional external data being provided after that point. By the end of self-training, the model is capable of accurately adding 30 digit numbers even without using chain-of-thought reasoning.

## 4 RELATED WORK

**Teaching Transformers Arithmetic.** While many attempts have been made to teach large language models to perform addition, to our knowledge, none have demonstrated that language models do so by teaching themselves. Lee et al. (2023) found significant challenges with length generation with transformers with very few parameters. Liu & Low (2023) fine-tuned the LLAMA family of models (Touvron et al., 2023), successfully teaching the models 8-digit addition using a supervised fine-tuning, while methods such as Zhou et al. (2022) try to teach addition via in-context learning instead

of fine-tuning. Nye et al. (2021) augments language models with the ability to emit intermediate computations to a temporary "scratchpad," allowing the model to perform up to 10-digit addition.

**Self-Learning in Language Models.** Prior approaches have been tried to perform self-learning in language models. For example, StAR (Zelikman et al., 2022) generates additional data for training by asking models to give rationales for a correct question-answer pair where none exist. It then trains again on those self-generated rationales and demonstrates improved performance on several reasoning-based benchmarks. Impossible distillation (Jung et al., 2023) takes an off-the-shelf LLM and then distills a high-quality dataset by filtering its generations. It then undergoes a self-distillation phase where it does the same process on its own generations, before undergoing one further phase of self-improvement. Huang et al. (2022) use a similar approach of chain-of-thought reasoning and self-consistency checks to improve at several reasoning datasets using very large models. However, the above methods fail to mitigate *error avalanching* in the training process, resulting in only a few steps of self-improvement before the process terminates.

Many other methods rely on self-improvement via methods other than updating the weights. Methods such as Voyager (Wang et al., 2023a) exploit the powerful in-context learning abilities of large language models to self-learn Minecraft abilities. Reflexion (Shinn et al., 2023) uses a similar approach to improve accuracy on the HumEval coding benchmark. SwiftSage (Lin et al., 2023) uses a similar idea of "fast-and-slow" thinking, but does not perform the self-learning loop of fine-tuning a model on its own generations. RAP (Hao et al., 2023) incorporates search into the decoding process. Self-debug (Chen et al., 2023) teaches a model to debug its own generated coding mistakes.

**Error Avalanching.** Much work has also been done on the difficulties of self-training, especially in language models. Shumailov et al. (2023) give both theoretical and empirical evidence that models repeatedly trained on their own outputs, without any filtering mechanisms, quickly degenerate into nonsense because of error avalanching. SECToR mitigates this by using chain-of-thought reasoning and self-consistency checks minimize the number of errors that accrue in the training process. Zhang et al. (2023b) coins the term *hallucination snowballing*, which can be viewed as a specific form of *error avalanching* in the context of generating factual content. Dziri et al. (2023) give some theoretical justification that errors may snowball exponentially. Wang et al. (2023b) used self-consistency to improve reasoning by sampling various independent methods of solving the problem before using a majority-vote system among the sampled solutions to construct the final answer.

## 5 DISCUSSION

We demonstrate that chain-of-thought reasoning can serve as a policy improvement operator and show a proof-of-concept demonstration that language models can teach themselves addition. In contrast to prior efforts in which self-improvement fails after only a few steps at most, models trained with SECToR manage to stay on pace for over twenty steps. Nevertheless, numerous avenues remain unexplored in the context of self-learning with language models.

**Limitations.** While SECToR demonstrates the possibility of self-learning in addition with language models, it is far from showing that models can self-learn in general. A natural question is whether methods like SECToR can generalize to more complex tasks, such as multiplication or perhaps even general mathematics or programming. Secondly, models trained with SECToR do not improve forever. We speculate that a larger model, or a stronger consistency check, might allow for the models to continue improving beyond 29 digits. Finally, while SECToR is data efficient, it is highly compute inefficient, requiring a large amount of compute to generate the next iteration's training data. We leave the development of more efficient methods for SECToR to future work.

**Safety.** While SECToR is merely a proof-of-concept demonstration of the possibility of self-learning in language models, this line of research brings both tremendous opportunities as well as potential risks. One risk is that self-learning may amplify preexisting biased or erroneous information in the model during the self-training loop. This is not a concern when considering purely objective domains such as addition, but may be an issue if self-learning is more broadly applied to other domains with less objectivity. Additionally, as models gain proficiency in autonomous learning, the boundaries of their capabilities may become less and less predictable, raising questions of how such models can be controlled and used in a safe manner. Alleviating these concerns is an important direction for future research.

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
