# OpenReview forum: "Chain-of-Thought Reasoning is a Policy Improvement Operator"
_ICLR.cc/2024/Conference — Submitted to ICLR 2024_

### Official Review · Reviewer_wTSJ · 2023-10-23

**Soundness:** 3 good
**Presentation:** 3 good
**Contribution:** 4 excellent
**Rating:** 8
**Confidence:** 5

**Summary:**

This paper shows that you can improve LLMs performance and LLM generalizability by a) first asking the LLM to show its work solving the same problem, effectively decomposing the problem into two smaller problem that the LLM has previously demonstrated proficiency on. They then use the solution obtained by chain of thought reasoning to train a new model that can solve the same problem without the decomposition step, and iteratively build up more powerful models.

They further address a number of useful technical issues in the implementation of this idea and how it differs from "standard" reinforcement and self supervised learning techniques. They explicitly discussing how to avoid error avalanching challenges using sanity checks and voting methods.

They demonstrate this using a standard and well chosen example of addition of integers. The generalizability in question is the ability of the model to scale up to more digits. In particular, they show that the self training methods can typically increase the generalizability by 5-10 digits.

**Strengths:**

In the dimensions of originality, quality, clarity and significance, this paper is strong.  Along those dimensions, there are three primary strengths to this paper:
1. The beautifully clear idea and and associated exposition. This paper makes it crystal clear what problem they are trying to solve and how they are going to solve it.
2. Excellent performance results. Being able to generalize from 7 to 29 digit numbers is a strong result
3. Their solution to error avalanches, which is a novel mix of meromorphic testing and voting/ensembling methods.

Taking each of the dimensions in turn.

## Originality
The idea of replacing a MCTS with a chain of thought is quite
novel. This is different from other approaches to generalization
because it does not rely on any quantity of data (e.g. semi-supervised
learning, data programming) instead using a generative model to
generate its own data, while exploiting specific capabilities of

## Quality

There were two points where the quality of this work was highlighted:
Three points where the quality of this work really came through were:
First, the detailed discussion of Error Avalanching in section 2.2 and
3.3.1 greatly improved this paper and has useful applications even
outside the LLM generalization (e.g. for testing). The second area was
the the supporting information was complete and well thought out. It
did a good job providing details and expanding on the results.

## Clarity
This had a clear statement of need and approach. I could clearly follow what they were trying to do, what the limitations of current practice were, and what was new in their approach.

## Significance
The use of addition is strong choice because this is an area where
LLMs are known to struggle. Work in addressing that is significant
automatically because there are huge gains to be made in the area.

**Weaknesses:**

## General comments:
- Stylistically, too much hyperbole as the results stand on their own. The discussion of alpha-* methods is curious but feels overblown. Most of the discussion is focused around a reinforcement learning perspective with the self learning and self teaching saved for the related work sections. It is likely a pedantic distinction, but some discussion of why this is reinforcement learning rather than semi-supervised learning would be useful. I believe the answer to this is that the methods here generate their own data rather than incorporate unlabeled data, but this distinction was hard to find in the discussion and felt drowned out by the hyperbole about alpha-*

- The results of this paper are very specific to the BYT5 models and the addition datasets. Those are very good examples for this paper, however the claims made are broader than a single model and a single dataset. In its current form, this a really a single method that works on a single dataset. A more compelling result would be multiple models and multiple forms of generalization.


## Originality
No weaknesses identified

## Quality

The primary weakness in the quality of the work and the experiments
was an over-reliance on a limited set of models and tests. Those tests
were well chosen, but in its current from, this is a single dataset,
single model proof of concept.



## Clarity
The related work seemed incomplete. In particular, the connections to
semi-supervised learning and other data efficent labeling methods
wasn't discussed but felt very relevant to the discussion.

The discussion could be significantly improved in two ways:
1. Some discussion of the specific results of this paper should be
   included. For instance, why is the generalization accuracy higher
   for 10-20 digit numbers than it is for 4-6 digit numbers?
2. The point about compute inefficency is a good one, but if it is
   included, specific references should be included for ways to
   improve the compute effiency would strengthen this section.

## Significance

The use of addition as a reference example is both a strength and a
weakness of this work. While I listed arithemetic above as a strength
because it is a known area where LLMs struggle, it is also a weakness
because pocket calculators have been able to add 30+ digit numbers
since they replaced slide rulers. That isn't meant to attack the
premise of this work, as I said, it is a strength, but the
over-reliance on a toy problem undercuts the significance of this
paper. The
claims made are about self teaching and generalization and those would
be better illustrated with a more broad cross section of problems, and
on problems that are not already solved by readily available, cheap
electronic devices such as calculators.

## Specific Concerns:
- page 7, paragraph 2. They claim "we use K=5, which was found to be a good balance between computational speed and accuracy". I think this is an important claim that greatly impacts their results, but no data is provided to support this claim.

**Questions:**

Questions are minimal and there are a number of specific questions

- Figure 6) This is a key figure for the paper and I have a number of questions about it:
1. I can't meaningfully read the accuracy from these bars.
2. There is better behavior in the middle than on the ends

- What are the key data efficient and curriculum learning references that you can use to improve these results?

---

> ### Author Response · Authors · 2023-11-20
>
> We thank the reviewer for their comments.
>
> **Regarding ablations on simplify-and-guess.** In Figure 5, we run the experiment measuring the impact of using K=0 (no decomposition and just chess) as well as K=N-1 (break the problem all the way down) compared to a middle ground of K=5. We will run an ablation measuring the impact of other intermediate values of K shortly.
>
> **Regarding Figure 6.** Thank you for pointing out the difficulty in reading accuracy in this figure. In the updated paper, we include a table of all accuracies in the Appendix for ease of reading. Our hypothesis for why there might be better generalization in the middle than at the ends is that the model may be reaching its capacity for learning (see general comment) as it allocates more and more of its “ability” to fast-adding larger and larger digit numbers.
>
> **Improving Data Efficiency.** In Appendix I, we find that doubling the parameter count of the model leads to almost a 10x reduction in data needed. An interesting  direction for future work is to see if this trend continues.

---

### Official Review · Reviewer_WpSN · 2023-10-29

**Soundness:** 2 fair
**Presentation:** 3 good
**Contribution:** 2 fair
**Rating:** 3
**Confidence:** 3

**Summary:**

The paper presents a proof-of-concept demonstration showing that language models can reinforce themselves to some degree through chain-of-thought (CoT) reasoning and fine-tuning on self-generated answers. The authors suggest utilizing the models' CoT capability as a policy improvement operator, similar to how Monte-Carlo Tree Search enhances AlphaZero. Experiments conducted on synthetic addition tasks reveal that small models, pretrained on data containing only a few digits, can self-learn to solve addition problems with up to 28 digits.

**Strengths:**

1. Linking CoT with Monte-Carlo Tree Search is an interesting and, as far as I know, novel idea. Exploring the direction of optimizing language models through self-training shows promise for model training.
2. The authors present intriguing observations on how error accumulation prevention enhances self-learning.
3. The paper is easy to follow.

**Weaknesses:**

1. Although the proposed method demonstrates strong self-learning abilities in synthetic addition tasks, I generally perceive it as being quite specific to those tasks. More specifically, the simplify-then-guess approach and curriculum appear to necessitate a distinct hierarchical structure that can be solved recursively.
2. The experimental results are not enough to confirm the feasibility of the proposed method. The absence of baseline methods raises concerns about whether the proposed method performs well or if it simply learns basic addition operators that are easy to learn. Additionally, I am unsure why the self-learning process fails at 29 digits. I recommend that the authors conduct a more comprehensive evaluation on the failure cases.
3. The authors propose hypotheses that, in my opinion, are inappropriate. For instance, the authors mention that "a sufficiently large pre-trained language model might be able to forgo the supervised training period entirely and begin self-training immediately, perhaps with only a few examples of in-context demonstrations." It would be safer to avoid phrasing it like this as there is no supporting evidence from relevant references or experimental results.
4. Some necessary details are missing. Please refer to the Questions part.

**Questions:**

1. How is the supervised training dataset generated? What distribution of prompts is used for answer generation during self-training? Approximately how many tokens are used for training in each phase?
2. How does curriculum learning contribute to final performance? I see no results in Appendix J.

---

> ### Author Response · Authors · 2023-11-20
>
> **General Applicability of SECTOR.** The reviewer expressed concerns about the general applicability of SECToR. In this paper, we merely wished to demonstrate a proof-of-concept for self-learning. However, we believe that the method is generally applicable to others tasks in which 1) models perform better after using chain-of-thought or another inference time decoding procedure, and 2) there is some form of self-consistency checking available.
>
> Recent research has suggested that many benchmarks are improved via chain-of-thought reasoning and related ideas. In terms of the self-consistency check, while the use of commutativity is specific to addition and related arithmetic operations, many other tasks have other forms of self-checking. For example, one could imagine proving a theorem in several different ways or navigating through a city via multiple different paths as a self-consistency check.
>
> **Supervised training Questions.** In Figure 2, we provide explicit examples of the data for both fast and slow addition. These examples can be easily generated programmatically and the script used to generate them is provided in the supplementary data in   generate_data.py.  N-digit addition problems are generated by randomly sampling two N-digit numbers. Appendix I provides a graph depicting how much data is used during the supervised/self-train period. In particular, Figure 10 shows that the 582M model used around 0.5M training examples during the supervised training period and self-generated an additional 1M training examples for supervised training. The number of tokens per example varies but is on the order of 100 for slow addition (per decomposition step) and 10 for fast addition.
>
> **Curriculum Learning.** Appendix J states that: “We run an ablation where we train a 582M parameter ByT5 model on 1 through 6 digit addition in a single supervised fine-tuning step, instead of via the curriculum learning setup done in Section 3.5. We find that this model, when properly trained, generalizes to up to 9 digit (slow) addition perfectly.” Our conclusion is that curriculum learning is not necessary in the  supervised training, but required in the self-training phase, as the model cannot generate the data for N+1 digit addition until it has learned to add N digit numbers. In the paper, we used curriculum learning in both phases because this provides that the unique difference between self-training and supervised training was the data generation.
>
> **Hypothesis on Emergence.** The reviewer took issue with our hypothesis that a sufficiently large LLM might be able to forgo supervised training entirely. This was framed as a hypothesis, as we did not have the resources to run this particular experiment. However, we politely disagree that there is no supporting evidence in the paper. Specifically, the preceding sentence refers to experiments done in Appendix I in which we found that that doubling the parameter count of the pretrained model (from 300M to 582M) reduced the training data needed in the supervised training period by almost a factor of 10 (from 4M to 0.5M supervised training examples). We did not have the computational resources to do an extensive study of scaling laws, but believe it is a reasonable hypothesis that a sufficiently large model might reduce to only a few examples. In particular, the SOTA LLMs have 1000x (~2^10) the parameter count of the models which we used here. Additionally, the history of LLMs are filled with tasks that previously required fine-tuning but can now be done with mere prompting (including chain-of-thought reasoning itself!). However, given this feedback, we have phrased our hypothesis more cautiously in the updated version of the paper.
>
> **Failure at 29 Digits.** In the general response, we described the results of a new ablation addressing this concern. This experiment sheds some light on the causes of the failure at 29 digits.

---

### Official Review · Reviewer_btm4 · 2023-11-01

**Soundness:** 3 good
**Presentation:** 3 good
**Contribution:** 3 good
**Rating:** 3
**Confidence:** 4

**Summary:**

The paper presents SECToR (Self-Education via Chain-of-Thought Reasoning) which helps language models to teach themselves new skills using chain-of-thought reasoning. SECToR contains two phases. The initial supervised fine-tuning phase employs curriculum learning to gradually introduce more complex addition problems. The self-training phase helps the model successfully generalize to N + 1 digit addition, even though it has been primarily trained on addition problems ranging from 1 to N digits, facilitated by chain-of-thought reasoning. SECToR also employs self-consistency and commutativity checks, to mitigate errors during self-training.

**Strengths:**

The article introduces the SECToR method, which achieves self-learning through chain-of-thought reasoning. This approach provides a new avenue for autonomous learning in language models.

**Weaknesses:**

1. This article lacks specific details regarding the dataset used, including the sizes of the training and testing sets, as well as the methodology for constructing the dataset (including both fast and slow datasets).

2. Figure 4 and the simplify-then-guess process is rather confusing. Can the authors provide a more detailed explanation or illustrative example of the simplify-then-guess process? (see question1,2).

3. The article should include a comparative experiment, specifically training the model directly on N+1 data to compare with the self-generated training data. The paper should also place greater emphasis on its motivation and advantages, as it appears that generating a large amount of additional data is a relatively simple and straightforward task. The current approach, in my opinion, may increase computational costs as it involves generating and verifying the data manually. Perhaps the article could explore applying this method to more challenging datasets beyond simple addition, where data acquisition is more challenging.

4. The article needs further improvement in terms of its presentation and image formatting. Some images are extending beyond the text width, which affects the overall layout and readability.

**Questions:**

1. In section 3.3.1, when saying “taking the first guess of a solution”, is the solution to the 8-digit problem or the 7-digit problem? If all the guesses belong to the simplified problem, how  the majority voting is conducted? Do the authors sample different answers for each sub-problem and conduct majority voting separately?

2. What is fast add? Does it mean generating the answer directly without using CoT (slow)?

3. This paper mentioned that "Training examples for each type are prefixed with a special token depicting their type, so the model can differentiate between the two." Could the authors please clarify what special token is being used for this purpose?

4. This paper mentioned that “SECToR checks that the answers generated by simplifying for one step, followed by immediately fast adding the subproblem emit identical answers for both a problem and its commutative twin.” I found the presentation somewhat confusing. Take figure 2 for example, do the authors do it by adding 4 and 14+12? If so, how to obtain the answer for 14+12?

---

> ### Author Response · Authors · 2023-11-20
>
> We thank the reviewer for their comments. Regarding the comments on formatting, these have been fixed in the updated version of the paper.
>
> **Fast adding / simplify-and-guess.** Fast adding indeed means generating an answer without using chain-of-thought. Figure 2 contains specific examples for both slow and fast addition. To describe simplify-and-guess, it is helpful to walk through Figure 4. Given an eight digit problem, the model first breaks down the problem into a seven digit problem (slow chain-of-thought), before taking a guess for that seven digit problem (fast addition). This is guess #1. To obtain further guesses, the model breaks down the 7 digit problem it produced in the previous step into a 6 digit problem, before taking a guess to obtain guess #2. This is repeated K times and all K guesses are aggregated via majority vote.
>
> Compared to simply decomposing the problem until it is 1-digit, simplify-and-guess is an additional error- checking mechanism in case there is error introduced in the decomposition step. Figure 5 describes the benefit that simplify-and-guess provides over just simplification or just guessing. Additionally, we hope to include an ablation explorating the impact of different choices of K shortly.
>
> **Data Prefix.** Training examples were prefixed with either “Add fast.” or “Add slow.” However, since the model was fine-tuned (and not just prompted) on these examples, the precise prefix is unlikely to be important since the model can easily learn the mapping between the prefix and the task at hand.
>
> **Regarding dataset details.** The reviewer claims that the paper lacks details about the dataset. We respectfully disagree. In Section 3.2, we state that “all training examples are generated programmatically by an external script during supervised learning.” This script is included in the supplementary data in the file called generate_data.py. In Figure 2, we provide explicit examples of the data for both fast and slow addition. Additional details about the dataset, including the precise size of the dataset and the split between validation and training, are given in Appendix B.
>
> **Ablation on Training on Oracle Data.** In the general response to reviewers, above, we provide the results of a new ablation that continues  the 29 digit self-training run, except with ground truth data after self-training has failed. As explained in the general response, we find that training can continue for a few more iterations, suggesting that SECToR’s self training performs worse, but not exceptionally worse than supervised fine-tuning with ground truth examples.

---

### Official Review · Reviewer_KjP6 · 2023-11-05

**Soundness:** 3 good
**Presentation:** 3 good
**Contribution:** 2 fair
**Rating:** 5
**Confidence:** 4

**Summary:**

The authors present a method for self-improvement of large language models via iterative chain-of-thought prompting and fine-tuning. Via this method they are able to produce a fine-tuned model that exceeds the current state of the art on zero-shot addition of numbers with large numbers of digits.

**Strengths:**

- The motivation for this paper is sound and strong. Self-improvement is likely to become an increasingly important area of research as data sources for training large models become exhausted.

- The results are well-presented, easily interpretable and improve on the current state-of-the-art in the particular domain under consideration.

**Weaknesses:**

- There are some missing citations e.g. in the self-improvement of LLMs space (https://arxiv.org/abs/2309.03409, https://arxiv.org/abs/2211.01910, https://arxiv.org/abs/2309.16797) and in the fine-tuning of LLMs via prompted LLMs space (https://arxiv.org/abs/2212.08410, https://arxiv.org/abs/2212.10071).

- My main concern is that this paper only demonstrates the benefit of the proposed method on a single toy domain: addition of numbers with many digits. Therefore, it is hard to assess whether this method can have more general impact for self-improvement. In particular, the use of commutativity in the method seems overfit to this domain, and wouldn't naturally generalize to a wider range of tasks.

- The paper does not make the setting precise until Section 3.3 (the italicised sentence). This framing should be provided much earlier in the paper, so as to avoid confusion for the reader.

- An important baseline is missing: namely what would happen if you kept training on the programmatically generated data with the curriculum that is used for the first phase of training? Can the authors comment on whether their method can outperform this baseline?

- Does the fine-tuned model lose the "general" capabilities that make it so valuable in the first place? The authors do not verify that fine-tuning does not reduce performance on standard LLM benchmarks. Can they comment on this?

- In Section 3.3.1 it is very unclear where the dataset of problems with N digits comes from. Is this generated by the large language model itself, or is this pre-generated synthetically? If the latter, the utility of the method is reduced, because the method is no longer self-contained. Could the authors clarify?

- In Section 3.3.1 it is unclear why the iterative method of "simplify then guess" is required. If the chain of thought prompt reduces N+1-digit addition to N digit addition, and the fast addition for N digits is highly reliable, then surely further decomposition is not required? It would be useful to see an ablation of the full "simplify then guess" method, compared with the simpler "one step" one which I have suggested here.

**Questions:**

See "Weaknesses".

---

> ### Author Response · Authors · 2023-11-20
>
> We thank the reviewer for their comments.
>
> **Regarding baselines.** The reviewer asks whether SECToR will outperform a baseline where the model is trained only on programmatically generated data. It will not. Our claim is not that self-training will perform better than training on ground truth data. We would not expect this behavior even if the data generation process has no errors, since in that case, it would only match and not exceed the baseline of training on ground truth data. Rather, our claim is  that self-training can occur in the absence of ground truth data. In the general response, above,  we describe a new  ablation in which we have replaced model-generated data with ground truth data after the self-training process fails (i.e., for digits 29+).
>
> **Regarding problem sources.** In SECToR, the problem statements themselves were provided to the model and not self-generated. In future work, we will explore asking the model to generate the problems themselves as well as the candidate solutions.
>
> **Regarding loss of existing capabilities.** We do not claim that SECToR produces an LLM that is superior to the original one -- rather, our goal is  to establish a proof-of-concept that self-learning in language models -- for multiple iterations that extend far beyond the original distribution of training -- is empirically possible. As per our general response, above, the paper’s primary goal is not to teach an LLM to perform addition — one could simply use supervised fine-tuning for this. The goal is to provide a proof of concept for self-learning, with  addition used as a testbed.
>
> **Regarding the importance of simplify-and-guess.** The reviewer’s proposed method would correspond to using simplify-and-guess with K=1 (simplify once then guess immediately). The summary is that although decomposition and fast-adding are fairly reliable, even a tiny error incurred in the data generation process is amplified in future iterations due to error avalanching. We will perform an ablation on the precise importance of K and include it in the next draft of our paper.
>
> **Regarding related work.** We don’t see much connection between the papers on distilling chain-of-thought utterances from a larger teacher model into a smaller student model and our  work, as we focus on self-learning where a model learns to self-improve without access to any external data source (whether from a larger teacher model or ground truth data). However, the other suggested related work is very relevant and included in section in the updated version of the paper with references to self-improvement via prompting.

---

> > ### Comment · Reviewer_KjP6 · 2023-11-22
> >
> > I thank the authors for their response. They have clarified some of the points which I found unclear. However the K=1 ablation does not yet seem to have been implemented in the latest version of the paper.
> >
> > Overall, I find the paper interesting, but my concerns about the relatively narrow scope of their task and the limitation of requiring an external source of problems remain. Therefore, I will maintain my current score.
> >
> > If the paper is not accepted, I hope that the authors will be able to extend their methods to a larger variety of domains in an updated version and resubmit to another top-tier conference. Best of luck!

---

### Author Response · Authors · 2023-11-20
**General Comment To All Reviewers**

We thank the reviewers for their responses. Since there was some overlap in the questions from reviewers, we first provide a general response directed to all reviewers.

**Usage of Addition as a Benchmark.** In this paper, our primary goal was not to propose a new method of teaching LMs how to add. The primary goal is to use addition as a proof-of-concept testbed that LMs can teach themselves new skills that are not mere interpolations of examples in the dataset. As such, our goal was to find a relatively simple task in which language models still struggled. Because past work has failed to produce successful self-learning results, we believe that it is important to first establish the possibility of self-learning on a simple task before moving onto more complex benchmarks.

Addition is well known to be a challenging task for language models. For example, several papers have shown that even enormous models like GPT-4 often struggle with addition (see the Teaching Transformers Arithmetic section of our related works). Now that the potential of self-learning in LLMs has been established, we plan to move onto more complex tasks for future work.

**Baseline Trained on Ground Truth.**  Reviewers requested a baseline comparison to a version of SECToR trained on ground truth examples. Relatedly, reviewers asked for additional analysis on why SECToR failed after 29 digits of training. This will be included in our updated version of our paper and summarized below.

In our main experiments and as described in the paper, the 582M model successfully completed training with 28 digits but failed to learn 29 digits satisfactorily. We have now completed a new experiment where we take the last successful checkpoint of the 582M model training run (at 28 digits) and continue the training run with only ground truth examples (instead of self-generated data). As such, error avalanching is no longer an issue since all data is guaranteed to be correct. When doing so, we observe the model can continue to learn successfully until again failing to train at 33 digits.

This experiment provides evidence that the failure of training (at 29 digits) is primarily due to accumulated errors / error avalanching, as hypothesized in the paper, but that the carrying capacity of the model (with respect to SECToR’s specific training procedure) is not too far away either and that either a different training method and/or a model with more parameters is needed to add even longer numbers. For future work, we will consider plotting the error rate of self-generated data with how many digits of addition the model can learn as a more fine-grained study of the above.

---

### Meta-Review · Area_Chair_tBjV · 2023-12-05

**Metareview:**

The paper introduces SECToR, demonstrating that language models can self-learn skills like adding large numbers using chain-of-thought reasoning and self-consistency for verification. Specifically, models initially trained on small numbers can autonomously learn to handle significantly larger digits, positing chain-of-thought as a policy improvement operator.

Nevertheless, the paper have some weaknesses that were not addressed by the authors, include:

- Limited Scope and Generalizability: The paper's focus is exclusively on the addition, which prompts questions about the SeCTor's applicability to more realistic or diverse tasks. Moreover, reviewers raised the concern that applicability of simplify and guess as well as use of commutativity might be restricted to a small set of problems. This narrow focus limits insights into how the SECToR approach might perform in different domains or with different types of problems. Ideally, applying SeCTor to a variety of tasks (e.g, such as GSM8K or MATH would make the paper more convincing.

- Retention of General Capabilities: The impact of fine-tuning on the model's general capabilities is not explored. It's unclear whether the fine-tuned model (on self-generated data) retains its broad utility or if the specialization in addition tasks comes at the cost of reduced performance in other capabilities. This could be explored even using a simple multi-task setup involving subtraction / multiplication.

Nit: The paper is using ICLR 2023 template, which might lead to desk rejection.

**Justification For Why Not Higher Score:**

As pointed by several reviewers, the paper's significance and contribution is somewhat limited due to its sole focus on the task of addition, potentially raising doubts about applicability of insights from this work beyond this narrow task.

**Justification For Why Not Lower Score:**

N/A

---

### Decision · Program_Chairs · 2024-01-16

Reject